# Hybrid ensemble 4DVar assimilation of stratospheric ozone using a global shallow water model

Douglas R. Allen, Karl W. Hoppel, David D. Kuhl

Remote Sensing Division, Naval Research Laboratory, Washington, DC, USA

*Correspondence to*: D. R. Allen (douglas.allen@nrl.navy.mil)

**Abstract.** Wind extraction from stratospheric ozone ($O_3$) assimilation is examined using a hybrid ensemble 4DVar shallow water model (SWM) system coupled to the tracer advection equation. Stratospheric radiance observations are simulated using global observations of the SWM fluid height ($Z$), while $O_3$ observations represent sampling by a typical polar-orbiting satellite. Four ensemble sizes were examined (25, 50, 100, and 1518 members), with the largest ensemble equal to the

number of dynamical state variables. The optimal length scale for ensemble localization was found by tuning an ensemble Kalman filter (EnKF). This scale was then used for localizing the ensemble covariances that were blended with conventional covariances in the hybrid 4DVar experiments. Both optimal length scale and optimal blending coefficient increase with ensemble size, with optimal blending coefficients varying from 0.2 to 0.5 for small ensembles to 0.5 to 1.0 for large ensembles. The hybrid system outperforms conventional 4DVar for all ensemble sizes, while for large ensembles the hybrid

produces similar results to the offline EnKF. Assimilating $O_3$ in addition to $Z$ benefits the winds in the hybrid system, with the fractional improvement in global vector wind increasing from ~35% with 25 and 50 members to ~50% with 1518 members. For the smallest ensembles (25 and 50 members), the hybrid 4DVar assimilation improves the zonal wind analysis over conventional 4DVar in the Northern Hemisphere (winter-like) region and also at the Equator, where $Z$ observations alone have difficulty constraining winds due to lack of geostrophy. For larger ensembles (100 and 1518 members), the

hybrid system results in both zonal and meridional wind error reductions, relative to 4DVar, across the globe.

## 1 Introduction

The extraction of wind information from stratospheric ozone ($O_3$) assimilation using a 4D data assimilation (DA) system is an attractive prospect, given the paucity of direct wind observations in the stratosphere. The tracer-wind relationship has been examined with a variety of DA systems including the extended Kalman Filter (EKF, Daley, 1995, 1996), 4D

Variational assimilation (4DVar, Riishøjgaard, 1996; Peuch et al., 2000; Andersson et al., 2007; Peubey and McNally, 2009; Semane et al., 2009; Han and McNally, 2010; Dragani and McNally, 2013; Allen et al., 2013, 2014), and Ensemble Kalman Filter (EnKF, Milewski and Bourqui, 2011, Allen et al., 2015). While idealized studies have shown strong potential for wind extraction from tracer assimilation, attempts to assimilate $O_3$ using realistic numerical weather prediction (NWP) systems have produced mixed results (see Allen et al., 2015 for a discussion). In an effort to understand the problem in more detail,

we previously developed a shallow water model (SWM) test case representing Northern Hemisphere (NH) winter stratosphere conditions. Assimilation experiments using both 4DVar (Allen et al., 2014; hereinafter A14) and EnKF (Allen et al., 2015; hereinafter, A15) showed that tracer assimilation is useful for wind extraction, but also raised issues such as sensitivity to measurement errors, localization, and choice of DA state variables, as well as the problem of imbalance.

Another approach to evaluating $O_3$-wind interaction in DA is to blend the 4DVar static covariance with flow-dependent ensemble covariance within the 4DVar. This hybrid 4DVar method is becoming increasingly popular at operational NWP centers (Buehner et al., 2010; Bonavita et al., 2012; Clayton et al., 2013; Kuhl et al., 2013; Kleist and Ide, 2015). In this paper, we extend our previous work by examining $O_3$-wind interactions using a hybrid 4DVar system within the SWM framework. Tuning of the length scale for the ensemble covariance localization as well as the covariance blending parameter are examined, in addition to probing the limits of wind extraction with a large ensemble experiment.

The layout of the paper is as follows. Section 2 describes the SWM hybrid 4DVar system and the experimental design. Section 3 describes hybrid results using both small and large ensembles, relative to the size of the state vector. Section 4 presents a discussion of the optimal assimilation experiments. Sections 5 and 6 provide a summary and conclusions, respectively.

## 2. Model description

### 2.1 Forecast model, truth run, and observations

The forecast model is the spectral SWM described in A14 and A15. For this paper, the model was run at a lower resolution of triangular truncation T21 (64 longitudes × 32 latitudes, for a Gaussian grid spacing of ~5.6° at the Equator) rather than T42 (which was used in A14 and A15) in order to facilitate a large number of tuning experiments and to allow the full background error covariance to be stored in active memory. To accommodate the lower resolution, the horizontal fourth order diffusion coefficient was increased from $5.0 \times 10^{15}$ $m^4$ $s^{-1}$ to $8.9 \times 10^{16}$ $m^4$ $s^{-1}$, which maintains an e-folding damping for the highest wavenumber of approximately one day. Other settings are the same as in A15, including a 10 km global mean height and time step of 120 s.

The truth run (TR) is similar to that used in A14 and A15. The system is initialized with a zonal jet with maximum wind of 60 ms$^{-1}$ in the NH, which is in geostrophic balance with the fluid height. A time-dependent topographic forcing is applied over the first 20 days. The shape of the forcing is the same as in A14 and A15 (i.e., a zonal wavenumber 1 mountain centered at 45°N), but the mountain height is increased from 1250 m to 1750 m to allow greater dynamical variability in the T21 system. After day 20, the topography is flat for the rest of the TR; since the assimilation begins on day 20, there is no

topography during the DA experiments. We use the same forecast model for the TR and the DA (i.e., "identical twin" experiments), making our results a best case scenario. The results are therefore likely to be overly optimistic.

Observations of $O_3$ and fluid height ($Z$) were generated by sampling the TR with the same frequency as in A15. The $O_3$ observations mimic Aura Microwave Limb Sounder sampling (one observation every 24.5 s), while $Z$ observations are pseudo-random in space and time, with the same sampling frequency. One change from A15 is that we increased the error standard deviations to 0.3 parts per million volume (ppmv) instead of 0.08 ppmv for $O_3$, and 200 m instead of 50 m for height. The 200 m error for $Z$ corresponds to approximately 1°K, using the scaling explanation in A15. Experiments assimilating either $Z$ only (referred to as "Z assimilation") or $Z$ and ozone (referred to as "Z/$O_3$ assimilation") are performed.

## 2.2 Ensemble Kalman filter

The EnKF is described in detail in A15. Briefly, it is a "perturbed observations" EnKF (Houtekamer and Mitchell, 1998; Evensen, 2003), with data assimilated in 20 minute batches. The EnKF analysis equation can be solved using different combinations of control variables. In this study, we use streamfunction, velocity potential, $Z$, and $O_3$ (the EnKF-$\psi\chi$ system), which was shown in A15 to have less imbalance than when zonal and meridional wind are used as the horizontal flow variables (also discussed in Kepert, 2009). To avoid filter divergence, we apply a state space covariance inflation factor (Anderson, 2007) to the background ensemble before assimilating observations. The inflation factor is designed to alter the global average ensemble spread in the streamfunction to match the global Root Mean Square Error (RMSE) in the streamfunction. We also apply the elementwise (Schur product) localization (e.g., Houtekamer and Mitchell, 2001) using Eq. (4.10) of Gaspari and Cohn (1999).

## 2.3 Hybrid 4DVar

The SWM 4DVar DA system is described in A14. The 4DVar minimizes a standard cost function using the accelerated representer approach (Xu et al., 2005; Rosmond and Xu, 2006) with a perfect model assumption. The conventional initial background error covariance $\mathbf{B}_0^{con}$ is calculated using an analytic formulation that employs wind-geopotential correlations based on approximate geostrophic balance on an *f*-plane, i.e., constant Coriolis parameter with latitude (Daley, 1991; Daley and Barker, 2001). There is no coupling between $O_3$ and dynamical variables in $\mathbf{B}_0^{con}$, but coupling between these variables does develop implicitly over the 4DVar time window. The background error standard deviations are adaptively tuned to match the globally averaged error standard deviations (with respect to the TR), as discussed in A14. The tangent linear model is also run at T21 resolution with the same diffusion coefficient and time step as in the nonlinear model. The 4DVar system runs with a 6-hour update cycle, and the 6-hour analysis at the end of one window is used to initialize the analysis at the start of the subsequent window.

For the horizontal correlation used in $\mathbf{B}_0^{\mathrm{con}}$, we found that the function used in A14 (single order auto regressive (SOAR) function with 1000 km length, sloping up to 1500 km in the tropics; see A14, Fig. 1a) was near optimal for the current experiments, even though the observations used in this study are much more sparse than in A14. Single observation experiments revealed that increasing the length scale in the $\mathbf{B}_0^{\mathrm{con}}$ introduces more gravity waves into the SWM system. The

imbalance is minimized with smaller length scales, since the *f*-plane assumption is more accurate. Reformulating the analytic balance for larger correlation lengths, or applying either a digital filter or nonlinear normal mode initialization within the variational solver, may further optimize the system. However, this is beyond the scope of the current paper.

To run the hybrid system (see Fig. 1 for schematic diagram), we first perform a 10-day EnKF simulation. We then run the

hybrid 4DVar over the same 10-day period in which the ensemble covariance, $\mathbf{B}_0^{\mathrm{ens}} = \mathbf{X}'\mathbf{X}'^{\mathrm{T}} / (N_{\mathrm{ens}} - 1)$, is calculated at the start of each 6-hour window using the ensemble states $\mathbf{X}'$. The prime indicates perturbation from the ensemble mean, the superscript T indicates transpose, and $N_{\mathrm{ens}}$ indicates the ensemble size. The ensemble covariance is then blended together with $\mathbf{B}_0^{\mathrm{con}}$ using $\mathbf{B}_0^{\mathrm{hybrid}} = (1 - \alpha)\mathbf{B}_0^{\mathrm{con}} + \alpha\mathbf{S} \circ \mathbf{B}_0^{\mathrm{ens}}$. Here $\alpha$ is a blending coefficient between 0 and 1, $\mathbf{S}$ is the localization function, and the open circle indicates the Schur product. Using the offline EnKF facilitates running the

hybrid system with a range of parameters without having to compute the ensemble each time. Tests in which the EnKF is re-centered about the 4D-Var analysis at the beginning of each cycle, as it would be done in an operational setting, produce similar results.

The experimental design is similar to A14 and A15. The DA experiments begin 20 days into the TR (day 20, 0 h), with the

initial state defined as the TR state that is offset 6 h from the initial time (i.e., day 20, 6 h). This initial 6 h offset, or mismatch, between the TR and the initial background fields is the source of the initial background error. The initial wind errors range from ~2-3 ms$^{-1}$ in the extratropics to ~3-6 ms$^{-1}$ in the tropics (see thin black lines on Figs. 9 and 10). We then perform 10-day assimilation experiments and compare the final wind errors with the initial wind errors. To illustrate how the wind errors evolve with time, Fig. 2a shows the global RMSE of the zonal wind over a 10 day DA period (the meridional

wind shows a similar trend) for one of the hybrid 4DVar experiments. The wind error starts at ~3.3 ms$^{-1}$, but  drops rapidly over the first several days before leveling out near day 6, suggesting that the system is well spun-up after ~6 days of assimilation. Figure 2b shows the Wind Extraction Potential (WEP), which is a normalized diagnostic relating the analyzed RMSE of the vector wind to the initial RMSE of the vector wind (a WEP value of 100% indicates perfect winds, while 0% indicates no improvement). Details of the WEP calculation are provided in A15. One slight difference is that in this study the

RMSE of the vector wind included an area-weighting factor that was not applied in Eq. (6) of A15. Figure 2b shows the

WEP starts at zero, but increases rapidly over the first several days, before leveling out at ~83% by day 10 of the assimilation experiment.

There are two main "tuning" parameters that we are considering for this study: the ensemble covariance localization length scale ($L$) and the hybrid blending coefficient ($\alpha$). While the localization length used in the hybrid blending does not have to match what is used in the EnKF, sensitivity tests showed that using the same length for both provides optimal or near-optimal results. Therefore in the experiments for this paper the same length is always used for both. Note that since inflation is automatically adjusted in a self-consistent manner with the TR, it does not require tuning.

The A15 paper quantified imbalance due to erroneous gravity wave modes that enter the EnKF system via imbalance in the analysis increments. Since the TR is virtually free of gravity waves due to the nature of the topographic forcing, any imbalance is considered to be unwanted noise. The imbalance can be reduced by judicious choice of flow variables and by tuning the localization length. In addition, A15 showed that application of nonlinear normal mode initialization (NMI) as a post-processing diagnostic improved the analysis in the EnKF system. In the NMI approach, the SWM state is first decomposed into three different mode types: eastward gravity waves, westward gravity waves, and rotational waves. The state is then adjusted using the Machenhauer (1977) condition, which reduces the time tendencies of the complex amplitudes of the gravity wave modes. We apply five iterations to solve the nonlinear balance equation using a cutoff frequency of 1.0 day$^{-1}$. This removes much of the imbalance and results in better agreement with the TR. For each of the experiments in this study (4DVar, EnKF, and hybrid), we therefore also compare results with and without NMI post-processing.

## 3. Results

### 3.1 Tuning the localization length

To examine the sensitivity of the 4DVar to the quality of the ensemble covariance, the offline EnKF is run at different ensemble sizes. Three "small" ensemble experiments are performed with 25, 50, and 100 members, and one "large" ensemble experiment is performed with 1518 members, which equals the number of degrees of freedom in the T21 SWM dynamical system (i.e., excluding ozone). The large ensemble experiment is used to explore the maximum benefit of ensemble covariance blending in the 4DVar system, while the small ensemble experiments test the performance of limited, or more practical, ensemble sizes. To initialize the small (large) ensembles, we sampled the TR at 6-h (36-min) intervals, starting at day 20. Experiments were also performed with 2024 members, which equals the total number of degrees of freedom in the system, including ozone. However, results were slightly worse than with 1518 members. This was likely caused by inadequate initialization of the large ensemble through sampling of the TR. Since the TR does not have topographic forcing, the system becomes less dynamically active as time progresses, and therefore the initial ensembles for the large runs may not be completely independent.

For the small ensembles, the tuning of the localization length was performed using 10-day EnKF experiments with a range of localization lengths, starting at 500 km and increasing in 500 km increments until the 10-day WEP values showed an obvious maximum. Due to intensive computation time, the large ensemble experiments were not finely tuned, rather localization lengths of 10,000, 15,000 and 20,000 km were tested. Tests with the large ensemble were also performed with no localization, but results were slightly worse. This may be caused by inadequacies of the initialization procedure, as explained above. Figure 3 shows the WEP as a function of length for eight combinations of observations ($Z$ or $Z/O_3$) and ensemble size (25, 50, 100, and 1518). Most experiments show smoothly-varying WEP as a function of length, with a well-defined peak. At 25 members the peak is narrow for $Z/O_3$, while for 100 members and $Z$ only, the peak is quite broad. The optimal lengths (i.e., producing maximum WEP) are indicated by vertical red lines; see Table 1 for numerical values. For the large ensembles, the WEP is not very sensitive to localization length, since the ensemble is sampling nearly all of the background error state.

One main conclusion from these tests is that both optimal length and optimal WEP increase with ensemble size. For small ensemble experiments, optimal lengths are also larger for the $Z$ only assimilation than for $Z/O_3$ assimilation. At 100 members, the optimal length of 14,000 km for $Z$ assimilation is quite long; this is likely due to the large-scale structure of the $Z$ fields in this experiment, combined with the relatively low resolution T21 system. Application of NMI increases the WEP for all ensemble sizes (see dotted lines in Fig. 3 and Table 1), with a larger impact on the $Z/O_3$ assimilation. This is consistent with A15, which showed that assimilation of $O_3$ tends to produce more gravity waves than $Z$ only, where there was very little imbalance. For the small ensemble experiments with $Z/O_3$, the optimal length scale also increases when NMI is applied.

## 3.2 Tuning the hybrid blending coefficient

We next tune the blending coefficient in the hybrid 4DVar system by performing 10-day experiments with values of $\alpha$ ranging from 0.0 to 1.0, in 0.1 increments, for each of the eight experiments. Figure 4 (top row) presents the WEP values for $Z$ assimilation as a function of $\alpha$ (NMI results are dotted lines). For each ensemble size, the optimal $\alpha$ is indicated (vertical red line) in addition to the range of $\alpha$ values that produce WEP within 0.5% of the maximum (vertical dashed lines). This range provides an indication of the flatness of the peak and the degree of flexibility in choosing $\alpha$. Plots of these ranges as a function of $\alpha$ are also provided in Fig. 5.

For $Z$ assimilation, the conventional 4DVar ($\alpha$=0) has WEP=67.6% (69.1% with NMI). For each ensemble size, WEP initially increases with $\alpha$, showing that ensemble covariances provide useful flow-of-the-day information in the system. The optimal blending coefficient for $Z$ assimilation increases from 0.1 for 25 members to 1.0 for 1518 members (see also

Fig. 5). The latter result indicates that for a "perfect" ensemble (i.e., one that samples the entire error space), the hybrid system benefits from using as much of the ensemble covariance information as possible. This is expected, since the large ensemble has the same number of degrees of freedom as the dyanamical state space of the SWM. The WEP values for $Z/O_3$ assimilation are provided in the bottom row of Fig. 4. The optimal blending coefficient varies from 0.2 for 50 members to 0.70 for 1518 members (see also Fig. 5). While the optimal blending coefficient increases monotonically for $Z$ assimilation, for $Z/O_3$ assimilation, the coefficient decreases from 25 to 50 members. The exact cause of this behavior is unknown, but it may be due to tuning the system with one length scale for both ozone and height. That the optimal $\alpha$ for 1518 members is not exactly unity suggests that the ensemble is not perfectly sampling the entire error space when $O_3$ is included in the state. While the WEP for conventional 4DVar ( $\alpha$ =0) is 78%, the peak hybrid WEP is ~86% for 100 members and ~89% for 1518 members. As it will be discussed below, the hybrid provides more benefit over 4DVar when $O_3$ is assimilated along with the $Z$, suggesting strong $O_3$-wind correlations.

Figure 6 shows the amount of imbalance entering the system for each experiment. Here we define "imbalance" as the global RMS (Root Mean Square) difference in $Z$ fields before and after NMI post-processing. For each ensemble size, the imbalance varies with $\alpha$ , with a minimum value that decreases with increasing ensemble size. Using more ensemble members therefore results in less imbalance in the system. The WEP values for $Z$ assimilation indicate only a slight improvement when NMI is applied (dotted lines in Fig. 4). The bulk of the improvements with ensemble size in the $Z$ assimilation experiments are likely due to more reliable information in the larger ensembles, rather than to reduced imbalance. As with $Z$ assimilation, the imbalance decreases with ensemble size for $Z/O_3$ assimilation (Fig. 6, bottom row). For 25 members, the imbalance increases monotonically with $\alpha$ , while for 1518 members the imbalance shows a minimum at $\alpha$ =0.7. There is a slightly larger benefit to applying NMI for $Z/O_3$ assimilation than for $Z$ assimilation (dotted lines in Fig. 4), particularly for small ensemble sizes.

## 4. Discussion of results with optimal tuning

We now compare the results from all three DA systems when using the optimal hybrid tuning parameters. Figure 7 shows the optimal 10-day WEP values for hybrid (blue) and EnKF (black) for all ensemble sizes and for 4DVar (red). The hybrid system outperforms the 4DVar, with WEP values increasing with ensemble size for both $Z$ and $Z/O_3$ assimilation. The hybrid outperforms the EnKF for small ensembles, while at 100 and 1518 members the results are similar. As ensemble size increases, it will likely become more difficult for the hybrid to beat the offline EnKF. There are therefore two limiting values of the hybrid system. The case of one ensemble member would be analogous to conventional 4DVar, while for large ensemble size the hybrid results are limited by the EnKF.

To quantify the added value of $O_3$ relative to the baseline system that assimilates $Z$ only, Fig. 8 shows the difference in global RMSE of the vector wind between the two sets of runs ($Z$ assimilation and $Z/O_3$ assimilation). The RMSE of the vector wind is calculated using Eq. (6) of A15, with the addition of an area-weighting factor. Note that larger positive numbers on Fig. 8 indicate smaller wind errors when adding $O_3$ to the system. The absolute difference (Fig. 8, left) shows that in the 4DVar system (red lines), $O_3$ reduces the wind error by ~0.32 ms$^{-1}$. In the hybrid system (blue lines), the wind error reduction is larger at 25, 100, and 1518 members, and similar at 50 members. With NMI applied (dotted lines), the $O_3$ benefit for the hybrid is larger than for 4DVar at all ensemble sizes. In Fig. 8 (right) the error reduction is given as a fractional reduction of the error when only $Z$ is assimilated. The reduction is ~30% for 4DVar, but increases to ~36-49% for the hybrid (and up to ~56% for 100 members with NMI). These results show that the added value of $O_3$ to the wind field is larger in the hybrid system than in 4DVar. This highlights the benefits of having initial $O_3$-wind covariances in the hybrid system that are not available in conventional 4DVar. The EnKF results (black lines) are also included in Fig. 8. Except for the large ensemble experiments, the relative benefit of adding $O_3$ is smaller in the EnKF system than in the hybrid. This suggests that the $O_3$-wind interaction benefits both from the ensemble covariances as well as the variational DA approach.

Lastly, we examine the wind errors as a function of latitude. Figure 9 shows initial zonal wind errors (thin black lines) along with final 4DVar (red), hybrid (blue), and EnKF (thick black) errors. All three systems show strong reductions from the initial errors. For $Z$ assimilation (top row), the 25-member hybrid shows a zonal wind improvement over 4DVar at high NH latitudes. Since the TR is topographically forced with a mountain centered at 45°N, this result is not surprising. With 50 or more members, the hybrid provides additional improvement in the tropics and parts of the Southern Hemisphere (SH). The hybrid $Z$ assimilation also reduces the meridional wind errors (Fig. 10, top row), ranging from modest NH improvements at 25 members to global improvements at 1518 members. The hybrid system shows generally smaller errors than the EnKF for 25 and 50 members, but results for these two DA systems are similar at 100 and 1518 members, as also shown in Fig. 7. Application of NMI does not alter the errors very much for $Z$ assimilation.

The zonal wind errors for $Z/O_3$ assimilation are plotted in Fig. 9 (bottom row). For 25 members, the hybrid system shows reduced errors in the NH and tropics, relative to 4DVar, while there are some slight increases in zonal wind errors near 30°S and 60°S. Why errors would increase at some latitudes when adding ensemble information is unclear, but it might be due to spurious correlations that are not localized. Since the optimization of the length is based on globally-averaged WEP, we might expect some regions to have increased errors. An ensemble localization scale that varies with latitude might be useful to consider here, but this is beyond the scope of this paper. As ensemble size increases, the hybrid errors decrease, although even at 100 members hybrid errors are still slightly larger than 4DVar errors at 60°S. At 1518 members, hybrid errors are smaller at all latitudes, and the tropical peak seen in the 4DVar errors is considerably reduced. This suggests that ensemble $Z$-wind correlations in the tropics are more reliable than the conventional correlations based on analytic balance assumptions.

The hybrid system shows generally smaller errors than the EnKF for Z/O$_3$ assimilation at 25, 50, and 100 members, but results are similar at 1518 members.

For meridional winds (Fig. 10, bottom), the hybrid system with Z/O$_3$ assimilation has smaller errors than 4DVar in the NH and in the midlatitude SH for small ensemble sizes, while at large ensemble sizes, the hybrid wind errors are smaller than 4DVar errors at all latitudes. The EnKF shows generally larger errors at small ensembles sizes than the hybrid, with a peak at over 2 ms$^{-1}$ near 20°S for 25 members. At large ensemble sizes the hybrid and EnKF results are similar. Including O$_3$ in the DA system reduces the overall wind errors, particularly in the tropics. This is consistent with the EnKF results in A15, which showed reductions in tropical errors when O$_3$ was assimilated. This is important, since Z observations alone have difficulty constraining the tropical winds, even in the hybrid system. But the O$_3$-wind tropical correlations have information that can reduce the wind errors there. We note as a caveat, however, that we have not yet attempted to include O$_3$ chemistry in the system, which may limit the tropical O$_3$ gradients, particularly in the middle and upper stratosphere. However, in the lower stratosphere, where the O$_3$ photochemical lifetime is long (except for ozone hole conditions), we might expect hybrid O$_3$ assimilation to provide a benefit particularly to the tropical winds. Chemistry is also very important for determining the ozone distribution in the lower stratosphere for polar ozone hole conditions. Finally, we note that application of NMI slightly reduces zonal and meridional wind errors for Z/O$_3$ assimilation for all three DA systems. For 25 members, the reduction occurs at nearly all latitudes, while for 1518 members the reduction is confined to the extratropics.

## 5. Summary

The problem of wind extraction from tracer observations in hybrid 4DVar data assimilation was examined in this study using a shallow water model (SWM) system coupled to an O$_3$ advection equation. While previous studies (A14 and A15) examined conventional 4DVar and EnKF simulations, this study combines the best of both systems by blending the ensemble covariance with the conventional covariance at the beginning of the variational assimilation window. The results show that O$_3$ provides added value in a system already constrained by height (in lieu of temperature for the SWM) observations, and that for small ensemble sizes, relative to the degrees of freedom in the state, the hybrid provides better results than either conventional 4DVar or the EnKF.

Using a relatively low resolution system, we were able to probe the limits of the benefit of hybrid covariance blending. Both the optimal localization length and the optimal blending parameter generally increased with ensemble size, so that with large ensemble size (spanning the dynamical state space), the optimal blending is essentially 1.0. For small ensembles (25 or 50 members), values of 0.2 to 0.5 produced better results. With large ensemble size, the hybrid system produced wind errors comparable with the offline EnKF, while for small ensemble size, the hybrid results were closer to 4DVar, suggesting the limiting benefits of hybrid blending. Overall, the hybrid outperforms the 4DVar, suggesting value in combining high-rank

conventional background error covariance with localized ensemble flow-of-the-day information when attempting wind extraction from tracers.

While wind extraction potential (WEP) was highest for large ensembles, even small ensembles provided information that benefited the hybrid system. We should note, however, that the ensemble size relative to the state vector is much larger than it would be currently possible for a full 3D NWP system. Therefore we hesitate to extrapolate the results to a full system. Another caveat is that the truth run we used was relatively smooth, due to the large-scale forcing applied. This may favor the 4DVar, since the tangent linear model (TLM) is likely to do quite well in this regime (see, for example, TLM errors in Fig. 3 of A14). Further tests with more complicated flows, such as the case of barotropic instability, would be valuable to examine the benefit of ensembles in highly nonlinear regimes.

The issue of balance also plays a role in DA with the SWM system. For small ensembles, imbalance generally increases as more ensemble information is added. When nonlinear normal mode initialization (NMI) is applied as a post-processing diagnostics, it benefits the winds in the hybrid system. The SWM, with minimal diffusion and no other physical parameterizations, is much more sensitive to imbalance than a typical operational system. How these results translate to operational systems is unclear, but at minimum they may provide some guidance as to when filtering (digital filter or NMI) may be useful.

## 6. Conclusions

The present work culminates a series of three papers (A14, A15, and the current work) examining the impact of tracer assimilation on winds using three modern operational data assimilation techniques (4DVar, EnKF, and hybrid 4DVar, respectively). The overarching goal of the tracer assimilation on winds is to use tracer data to fill the gaps in direct wind observations, particularly in the upper troposphere, stratosphere, and mesosphere. Since trace gas observations are not generally available at sufficient resolution for deriving feature-track winds, they must be combined with model background information to produce an analysis using a 4D data assimilation system. A pilot study using a full 3-D NWP model with a 4DVar system (Allen et al., 2013) showed that wind extraction from ozone assimilation in the stratosphere is possible, but also highlighted limitations due to geophysical conditions, tracer observation quality and error specifications, and limited observing sampling patterns. These results motivated the authors to pursue a more detailed theoretical study of the problem using the shallow water model (SWM) framework with a variety of data assimilation systems. Below we provide some overall reflections on the SWM experiments and suggestions for future work.

In the first SWM study (A14), we examined the relative benefit of assimilation of different tracers (ozone, nitrous oxide, and water vapor) in a conventional 4DVar system. Since the conventional 4DVar does not have correlations between tracers and

wind in the initial background error covariances, the only way that tracer assimilation can affect the winds is through the adjoint of the tangent linear model (TLM), which propagates sensitivities of the cost function with respect to tracer observations backwards in time. This approach is effective, as long as the background error covariances are correctly modeled. The 4DVar system has an advantage over the ensemble methods in that imbalance appears to play only a minor role. The analysis increments also tend to be smoother than when ensembles are used. One important conclusion of this study was that wind extraction will be easier with certain tracer characteristics (e.g., large background gradients and small observation error standard deviations).

The second SWM study (A15), which took the EnKF approach, illustrated the benefits of using ensemble correlations to propagate information from ozone observations to the dynamical variables. This proved to be very effective for extracting wind information from ozone, even in a relatively data-rich environment. The issue of spurious gravity waves played a larger role in EnKF than in 4DVar, due to imbalance caused when localizing the covariance. Imbalance was shown to be reduced by judicious choice of variables and increased ensemble size. However, imbalance is still an issue that needs to be studied further in the context of tracer assimilation using ensemble methods.

The current study showed that the largest benefit to the winds from ozone assimilation occurs in the hybrid 4DVar, which combines the benefits of variational DA with flow-of-the-day covariances generated from ensembles. While imbalance is still an issue, the blending of ensemble covariance with conventional covariance reduces the generation of gravity waves. The additional tuning required in the hybrid system does somewhat limit the applicability of the method. However, much of the tuning (e.g., localization) will already be performed in development of the 4DVar and EnKF (or other ensemble approach) systems. The main additional requirement is the tuning of the blending coefficient, which can be coarsely done if the sensitivity is small. The overall conclusion is that hybrid 4DVar offers the most promising approach (of the three DA methods we examined) for tracer-wind extraction in NWP.

Several future directions are being considered for this work with the SWM system. We would like to examine the effects of $O_3$ photochemistry on the wind extraction. It is likely that when the photochemical lifetime is short, the ability to extract wind from $O_3$ may be reduced, since advection is not the dominant term on the $O_3$ continuity equation. We would also like to test how this limited resolution study scales upward to more realistic systems. Further work is needed in separate tuning of the localization lengths for different variables, as well as filtering out gravity waves within the hybrid 4DVar system. In addition, we would like to explore eliminating the tangent linear model and adjoint using different ensemble variational approaches (e.g., Buehner et al., 2010, 2013, 2015; Lorenc et al. 2015; Frolov and Bishop, 2016). Finally, we plan to apply what we have learned from these SWM studies to devise experiments with an operational hybrid 4DVar system, building on the work of Allen et al. (2013), who examined wind benefits from $O_3$ assimilation in a pre-operational version of the Navy Global Environmental Model 4DVar system.

## Acknowledgments

We would like to thank Thomas Milewski, one anonymous reviewer, and the editor for helpful comments on the manuscript. Douglas R. Allen and Karl W. Hoppel acknowledge support from Office of Naval Research base funding via Task BE-033-02-42. David D. Kuhl and Karl W. Hoppel acknowledge support from Office of Naval Research base funding via Task BE-435-050.

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

| Experiment | $L$ (km) | WEP-EnKF (%) | WEP-4DVar (%) | WEP-Hybrid (%) | $\alpha$ (unitless) |
|---|---|---|---|---|---|
| Z | | | 67.6 (69.1) | | |
| 25 members | 6000 (6000) | 62.5 (63.9) | | 70.0 (71.0) | 0.1 (0.1) |
| 50 members | 7500 (7500) | 70.5 (71.1) | | 72.5 (73.8) | 0.4 (0.9) |
| 100 members | 14000 (13500) | 75.3 (75.7) | | 75.5 (76.5) | 0.8 (0.8) |
| 1518 members | 15000 (15000) | 77.5 (77.8) | | 78.3 (79.3) | 1.0 (1.0) |
| Z/O$_3$ | | | 77.1 (78.7) | | |
| 25 members | 3500 (5000) | 73.9 (79.1) | | 81.3 (84.8) | 0.3 (0.5) |
| 50 members | 5500 (6000) | 78.3 (81.8) | | 82.2 (84.8) | 0.2 (0.4) |
| 100 members | 7500 (8000) | 84.9 (87.8) | | 86.3 (90.0) | 0.5 (0.5) |
| 1518 members | 20000 (20000) | 89.2 (90.3) | | 89.1 (90.7) | 0.7 (1.0) |

**Table 1.** Results for the optimal runs (i.e., maximum wind extraction potential (WEP), in %), for each experiment. The localization length ($L$) is provided along with WEP. Results with NMI applied as post-processing of the analysis fields are provided in parentheses.

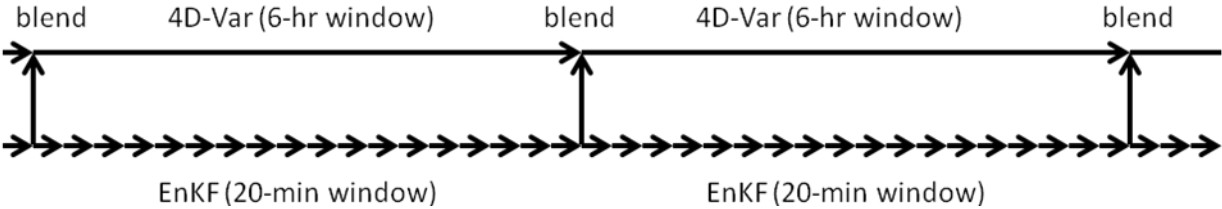

**Fig. 1.** Schematic diagram of hybrid system. The 4DVar uses a 6-h window, while the offline EnKF uses a 20-min window. At the beginning of the analysis window, information is passed from the EnKF to 4DVar by blending the covariances.

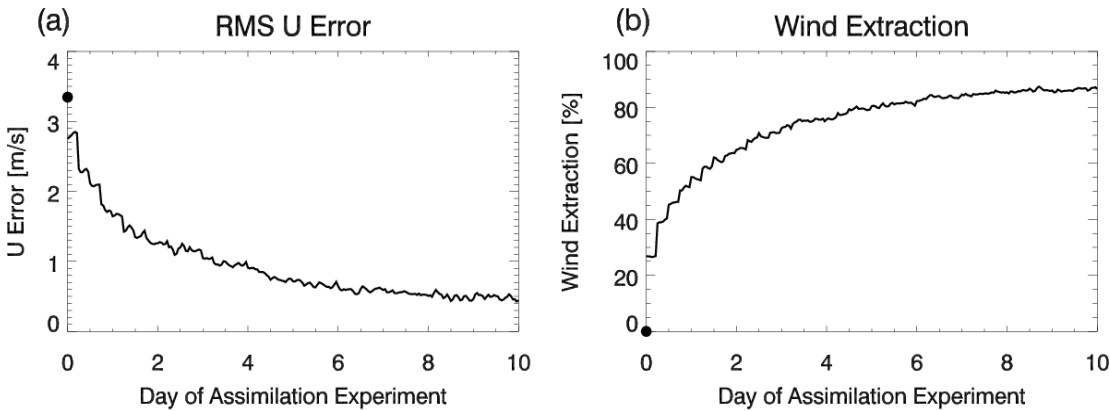

**Fig. 2.** (a) Global RMSE of the zonal wind (in ms$^{-1}$), and (b) Wind Extraction Potential (in %) for the optimal 100 member hybrid 4DVar for $Z$ /$O_3$ assimilation. Black circles indicate the values at the start of the assimilation experiment.

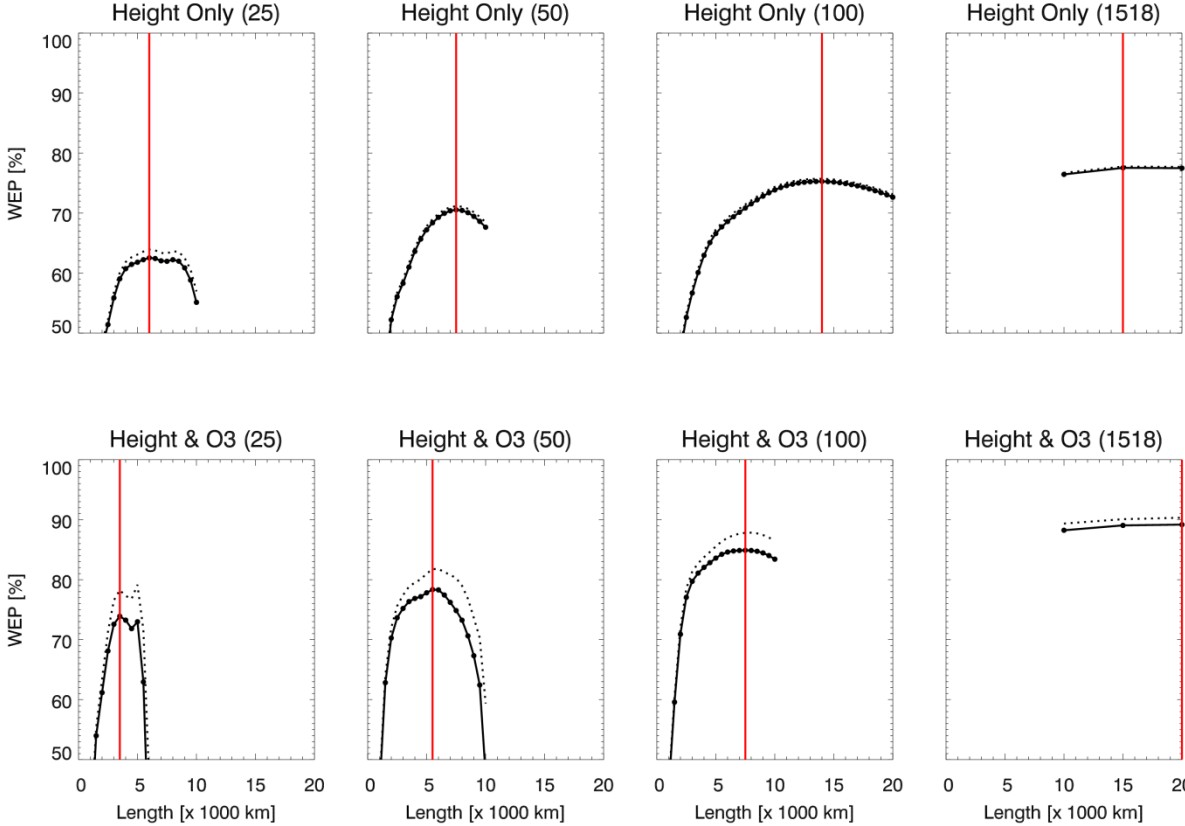

**Fig. 3.** Wind extraction potential (WEP, in %) as a function of localization length scale calculated from offline EnKF experiments assimilating $Z$ (top row) and $Z/O_3$ (bottom row) for ensembles with 25, 50, 100, and 1500 members (columns 1, 2, 3, and 4, respectively). Solid (dotted) lines indicate results without (with) NMI post-processing. Vertical red lines denote the length scale that resulted in the maximum WEP (without NMI post-processing).

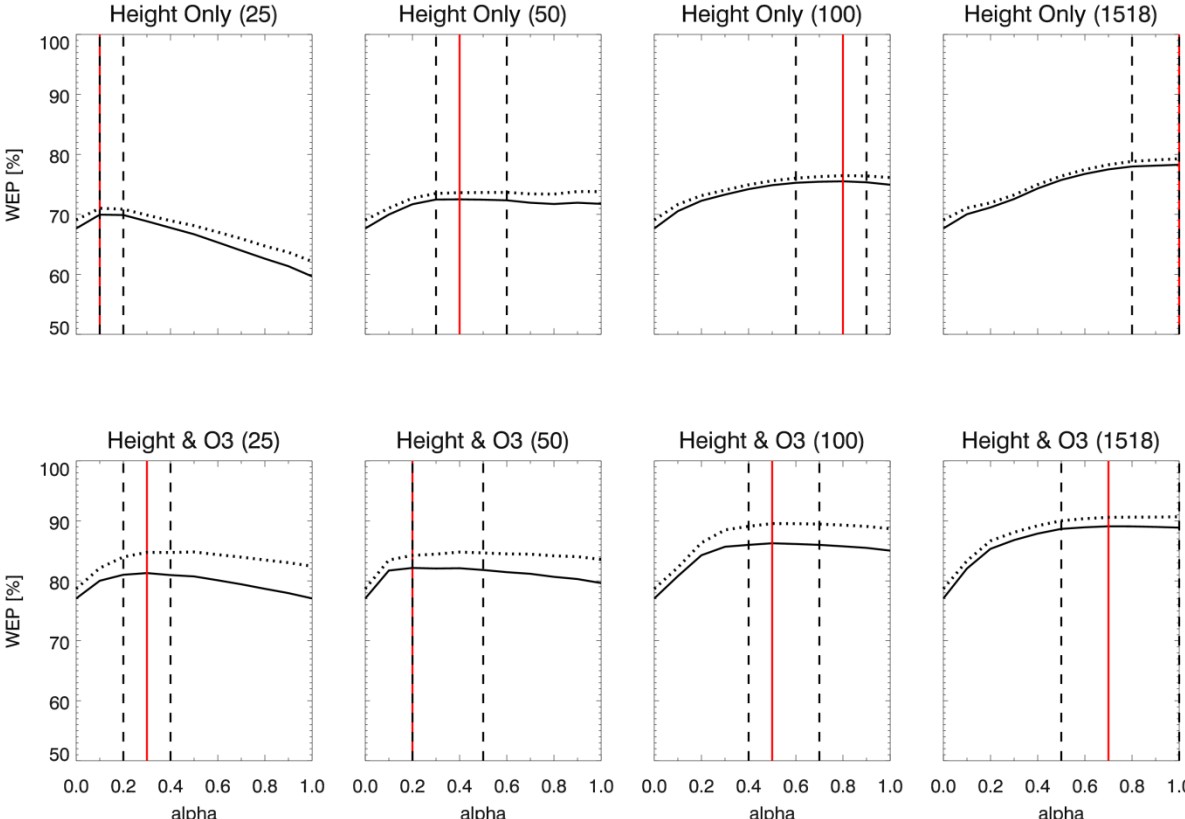

**Fig. 4.** Wind extraction potential (in %) versus blending coefficient for $Z$ assimilation (top row) and $Z/O_3$ assimilation (bottom row) for ensembles with 25, 50, 100, and 1500 members (columns 1, 2, 3, and 4, respectively). Solid (dotted) lines indicate results without (with) NMI post-processing. Vertical red lines denote the blending coefficients that resulted in the maximum WEP (without NMI post-processing). Vertical dashed lines indicate range of blending coefficients that resulted in WEP values within 0.5% of the maximum.

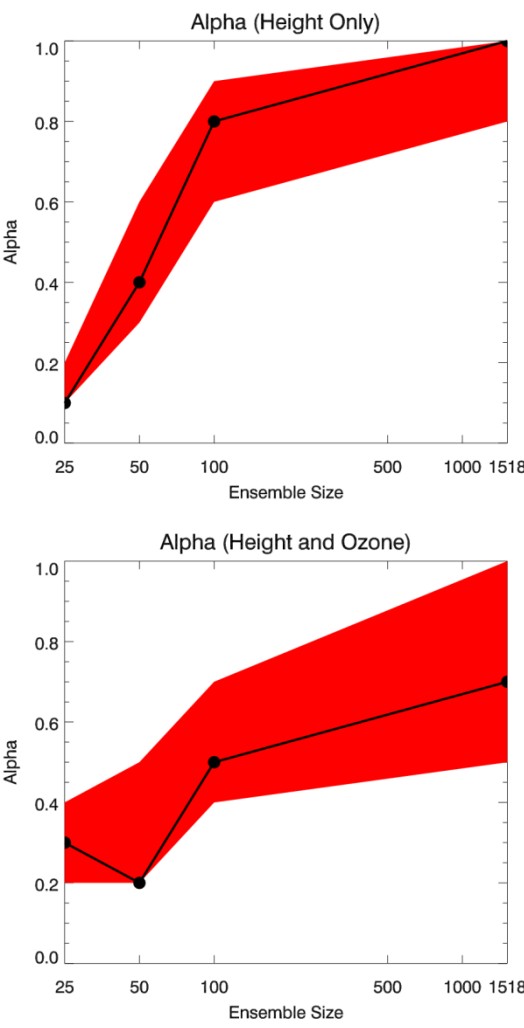

**Fig. 5.** Black circles indicate optimal blending coefficient, $\alpha$ (unitless), as a function of ensemble size for $Z$ assimilation (top) and $Z/O_3$ assimilation (bottom). The range values in red indicate hybrid experiments with WEP values within 0.5% of the maximum WEP for each ensemble size.

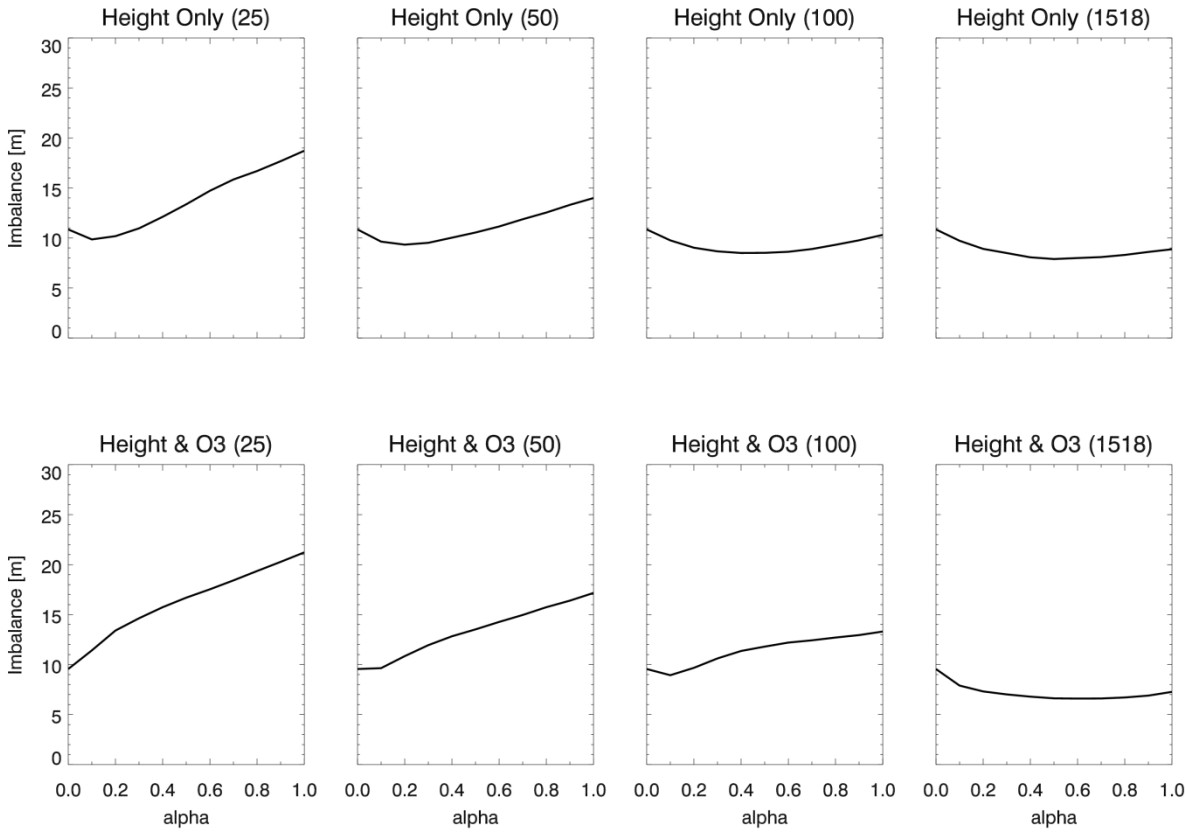

**Fig. 6.** Imbalance (in m) versus blending coefficient for $Z$ assimilation (top row) and $Z/O_3$ assimilation (bottom row) for ensembles with 25, 50, 100, and 1500 members (columns 1, 2, 3, and 4, respectively).

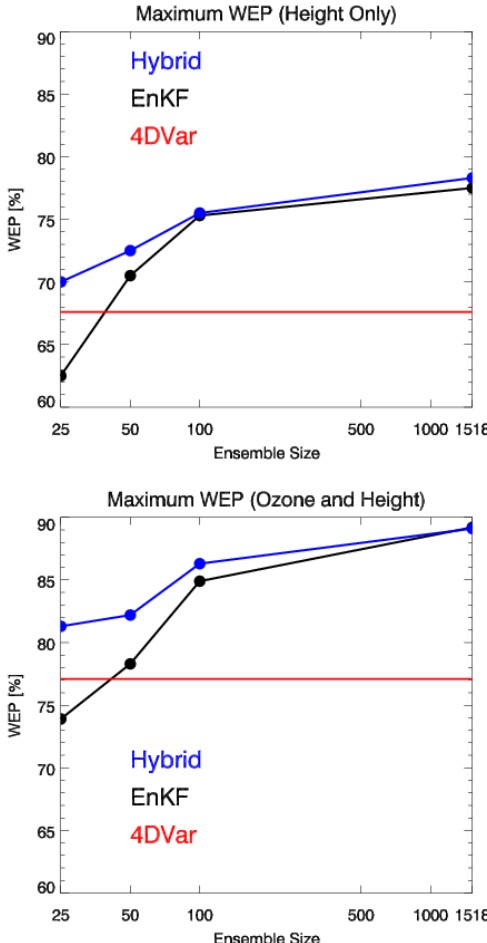

**Fig. 7.** Solid circles with connecting lines indicate maximum WEP (in %) as function of ensemble size for $Z$ assimilation (top) and $Z/O_3$ (bottom) assimilation. WEP is shown for EnKF (black), and hybrid (blue). Red lines indicate the maximum

5    WEP for 4DVar.

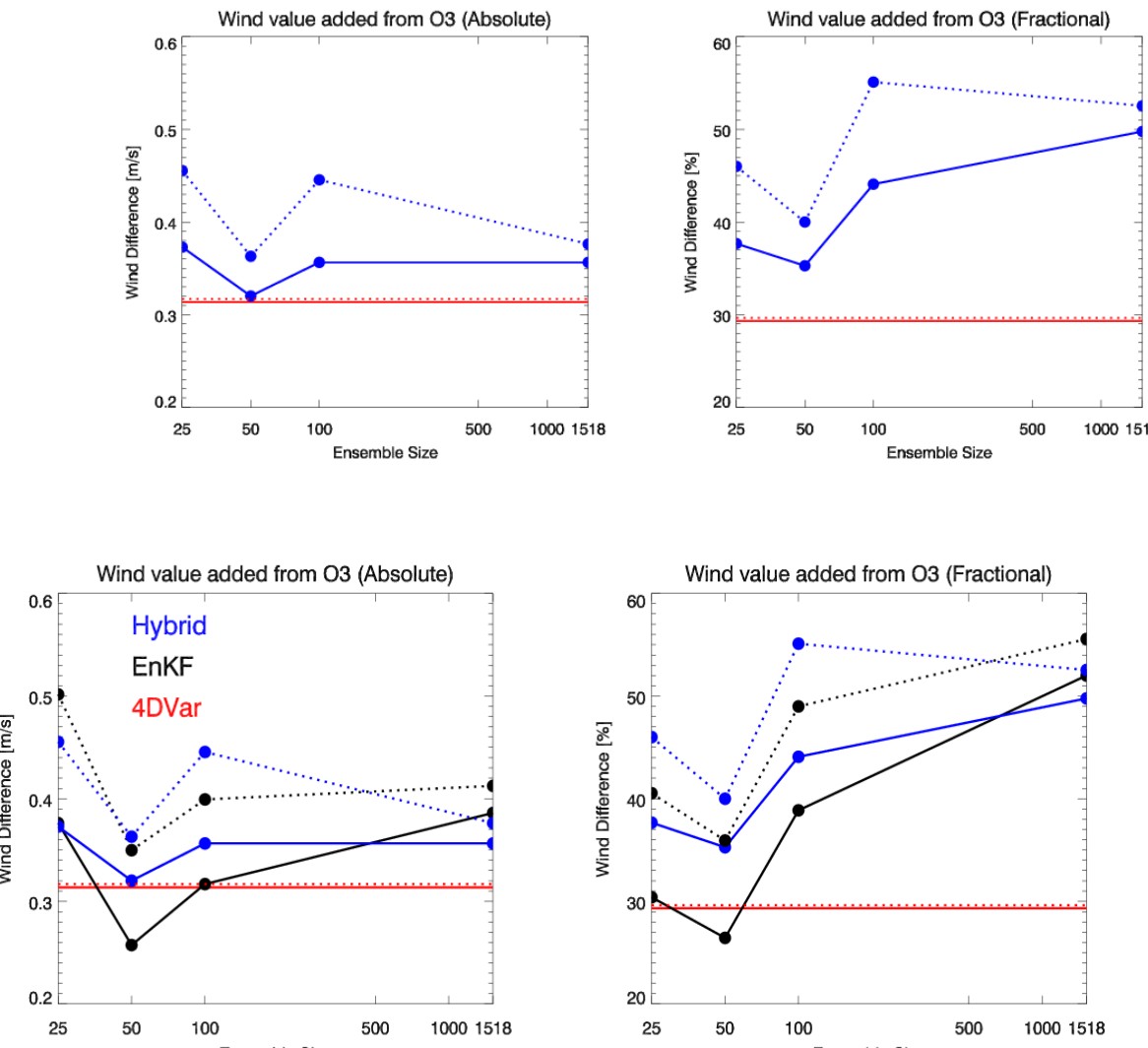

**Fig. 8.** Differences in the global RMSE of the vector wind between experiments that assimilate $Z$ only and experiments that assimilate both $Z$ and $O_3$. For each ensemble size, the final RMSE of the vector wind is calculated for each experiment and then the differences are taken. Left: the absolute difference in $ms^{-1}$. Right: the fractional difference (i.e., the difference from (a) divided by the RMSE of the vector wind for $Z$ only) in %. Red is 4DVar, blue is hybrid, and black is EnKF. Solid circles indicate the values for each ensemble, while solid (dotted) connecting lines differentiate results without (with) NMI post-processing. Note that larger positive numbers indicate smaller errors due to adding $O_3$.

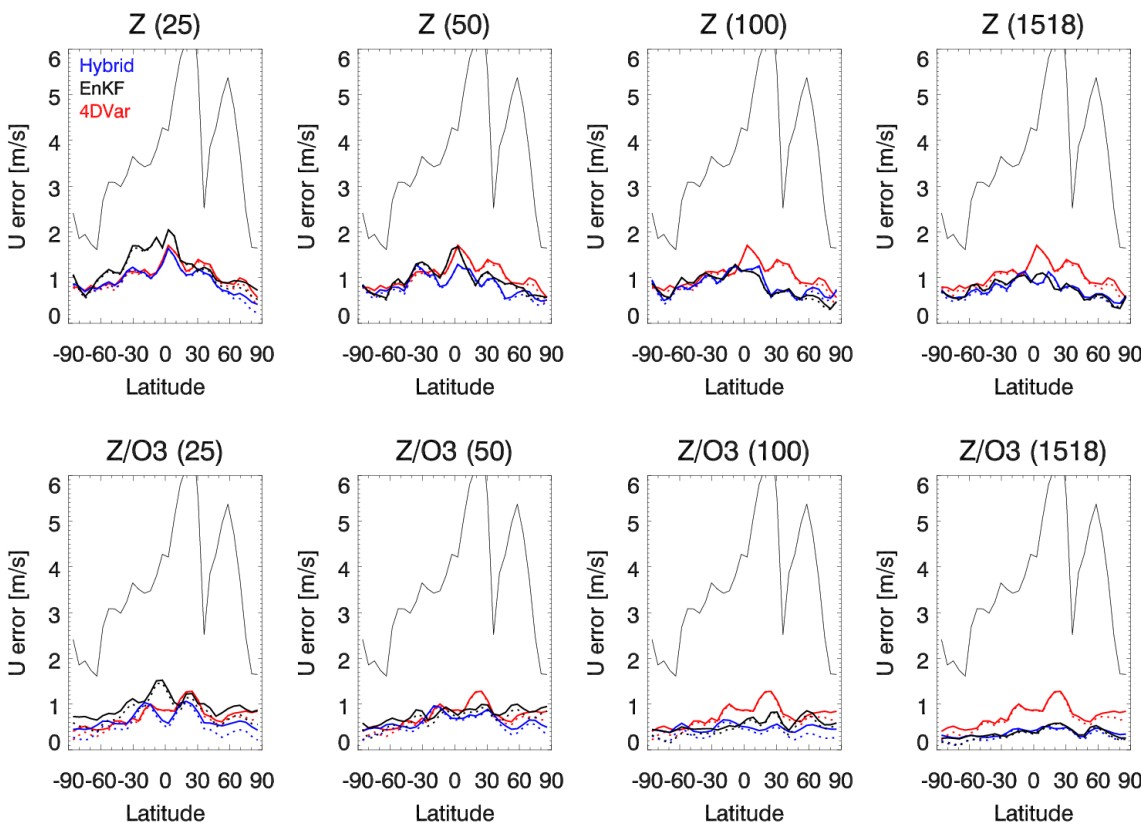

**Fig. 9.** Zonal wind errors (in ms$^{-1}$) as a function of latitude for initial conditions (thin black), and final results for hybrid (blue), EnKF (thick black), and 4DVar (red) for $Z$ assimilation (top row) and $Z/O_3$ assimilation (bottom row) for ensembles with 25, 50, 100, and 1500 members (columns 1, 2, 3, and 4, respectively). Solid (dotted) lines indicate results without (with) NMI post-processing.

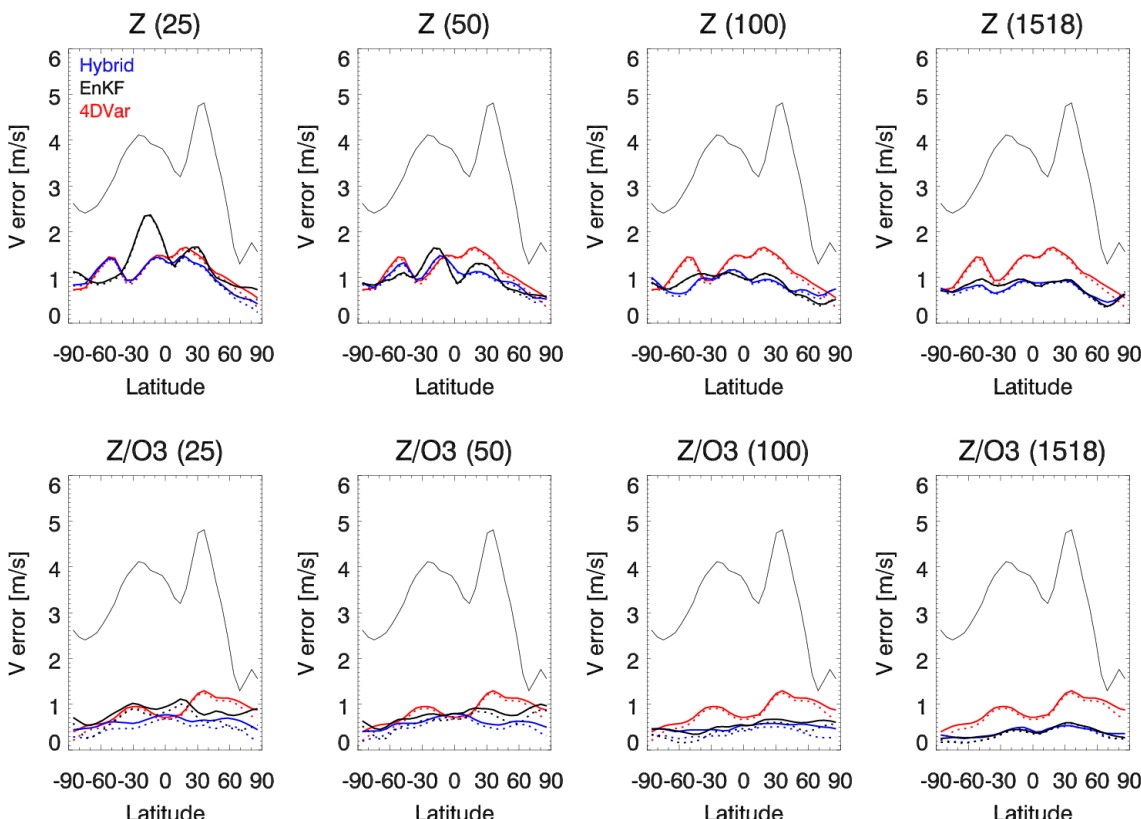

**Fig. 10.** Same as Fig. 9, but for meridional wind.