# Peer review of "Hybrid ensemble 4DVar assimilation of stratospheric ozone using a global shallow water model"

_Atmospheric Chemistry and Physics, 2016_

## Referee Comment (RC1)

**Review of the paper "Hybrid ensemble 4DVar assimilation of stratospheric ozone using a global shallow water model" by D. R. Allen et al.**

**General comments:**

This paper presents an assessment of the ability of a hybrid 4D-Var/EnKF data assimilation system to extract wind information from the assimilation of stratospheric ozone data, and compares it with that of a standard 4D-Var and a standard EnKF. Understanding if and to what extent ozone observations can constrain the stratospheric wind field in a data assimilation system is of a particular interest considering the scarcity of stratospheric wind measurements.

I find the topic very interesting, and within the profile of this journal. The quality of the study is high, the paper is clearly written and structured, and the results are discussed thoroughly.

I recommend the publication of this manuscript in ACP subject to a few minor revisions and comments (see below).

**Specific comments:**

1. Page 2 Lines 29-31 (P2L29-31 hereafter): I would have been more comfortable if two different models were used for the TR and DA. As it has been done, the DA experiments are constrained "twice" with the same information (the first time through the background that comes from using in the DA runs the TR model, and the second time by assimilating pseudo-observations derived from the TR itself). I appreciate the statement "results are therefore likely to be overly optimistic", and the care the authors put in designing the experiments, for instance by providing an initial error in the background resulting from imposing the Day=20 T=6h TR fields to be the initial condition for the DA, but I do wonder how significant the differences between the Day=20 T=0 and Day=20 T=6h TR fields really are. This could perhaps be commented in the paper.

2. Related to point 1 above: How responsive is the system? This aspect can help the interpretation of the results on two fronts: *a)* to disentangle the impact of essentially using the same constraint twice (in the sense mentioned above) and possibly quantify the actual wind extraction potential of ozone data assimilation (i.e. understand how overly optimistic the results are); and *b)* to know if at day 10 the system is completely spun up. A comment on this point should perhaps be included as well.

3. Section 2.2 and P5L20: In my understanding, inflation and localization are used in EnKF to address issues (such as filter divergence or long range spurious correlations) that result from under-sampling, i.e. having an ensemble so small in size that is not statistically representative of the state of the system. I'd argue that this does not apply to the case of the "large" ensemble with 1518 members and it would also explain why for that ensemble size the WEP value does not significantly change as function of the localization length (figure 2). The authors may want to comment on this and explain why and in what sense the large ensemble results that used no localization were worse than those with localization (P5L20).

4. P5L4-6: I appreciate the reference to A15, but perhaps the authors can say something more here on the NNMI and make this manuscript more self-contained. Also in A15, instead of NNMI the authors used the acronym NMI. It would be good perhaps to either use a consistent acronym or acknowledge the use of a different name.

5. P7L11-12: I guess the wind vector RMSE is computed as in equation 6 of A15, perhaps a reference can be added.

6. Fig 7: You may want to specify in the right panel caption how the relative difference is computed. I find interesting the reduced efficiency of the 50-member ensemble compared with those with 25 and 100 members. Is the reason for such a result known?

---

## Referee Comment (RC2) · T. Milewski (Referee) · 22 May 2016

General notes

This manuscript addresses the outstanding issue of wind extraction from the assimilation of ozone (O3) in an advanced data assimilation system: a hybrid ensemble 4DVar. This paper is a logical follow-up on its parent studies that looked at wind extraction from O3 assimilation in a 4DVar system (A14) and an EnKF system (A15). The authors studied the issue in a simplified yet properly-constructed framework and appropriately described the limits of the experimental setup. The results are interesting and instructive on the ability of this advanced data assimilation system of reconstructing wind analyses in the absence of wind observations, versus the more standard 4DVar and EnKF data assimilation systems. Based on these considerations, we recommend

that this paper be published in ACP following minor revisions.

Specific notes

1) The choice of using streamfunction and velocity potential as control variables seems to be based on the experiments ran in A15 with the EnKF. What about the impact of choosing this set of control variables versus zonal and meridional winds in the 4Dvar system?

2) How did the authors come up with the value of 1518 as the number of dynamical state variables in the T21 system? If I understand correctly, this is roughly the number of degrees of freedom in the dynamical system, but does it include the influence of O3? If not, this might explain why the optimal blending factor was not 1.0 for the large-ensemble Z/O3 assimilation (before NNMI).

3) Optimal localization lengths for Z-only and Z/O3 assimilation are very different, suggesting that O3 error covariance structures and Z error covariance structures are probably different. Ideally, this should require separate localization lengths. Did you try tuning Z and O3 localization lengths separately?

4) In figures 7, 8 and 9, the hybrid system is only compared to 4DVar, not EnKF. Since EnKF seems to outperform 4DVar for moderate to large ensemble sizes (at least in terms of WEP), it would be instructive to see the improvement that the hybrid system brings with respect to second-best performing system.

5) Considering that this paper seems to complete a trilogy on the topic of wind extraction in a hierarchy of data assimilation systems, it would be interesting to have a final paragraph in the "Conclusions" section that is a more extensive review of the behavior of the different DA systems, possibly including the pros and cons of each.

Technical notes

1) P.6 L.16: Please correct "The WEP value for of Z/O3"

---

## Author Comment (AC1) · 9 Jun 2016

**Reviewer comments in black, our responses in red. Revised paper with tracked changes is appended after our responses.**

**Response to RC1 from Anonymous Referee #2, 31 March 2016**

**Specific comments:**

1. Page 2 Lines 29-31 (P2L29-31 hereafter): I would have been more comfortable if two different models were used for the TR and DA. As it has been done, the DA experiments are constrained "twice" with the same information (the first time through the background that comes from using in the DA runs the TR model, and the second time by assimilating pseudo-observations derived from the TR itself).

We appreciate the reviewer's comments regarding the "identical twin" approach used in this study. This study, along with the previous two papers, was designed to lay the groundwork for understanding the interactions of tracers and winds in the data assimilation framework. Given the limitations of the shallow water model as an approximation to the full 3-D atmosphere, we do not expect quantitative results to be applicable to the real world. However, our hope is that these studies will benefit future, more realistic, studies.

I appreciate the statement "results are therefore likely to be overly optimistic", and the care the authors put in designing the experiments, for instance by providing an initial error in the background resulting from imposing the Day=20 T=6h TR fields to be the initial condition for the DA, but I do wonder how significant the differences between the Day=20 T=0 and Day=20 T=6h TR fields really are. This could perhaps be commented in the paper.

The initial zonal and meridional wind errors as a function of latitude are shown by the black lines on newly numbered Fig. 9 and Fig. 10, respectively. Since these aren't referenced until much later in the paper, we decided to add a brief discussion of the initial error in Sec 2.3. We also added a time series plot of the global RMSE of the zonal wind and WEP (new Fig. 2) as described further below, which addresses the issue of spin-up.

2. Related to point 1 above: How responsive is the system? This aspect can help the interpretation of the results on two fronts: *a)* to disentangle the impact of essentially using the same constraint twice (in the sense mentioned above) and possibly quantify the actual wind extraction potential of ozone data assimilation (i.e. understand how overly optimistic the results are); and *b)* to know if at day 10 the system is completely spun up. A comment on this point should perhaps be included as well.

We included in the revision a time series plots of the global RMSE of the zonal wind and WEP (new Fig. 2) to show how the system evolves. This figure is for the optimal results with hybrid 4DVar assimilation of height and ozone using ensemble size of 100 members. This shows that the system is fairly well spun up after about 6 days. The last two figures (newly numbered as Fig. 9 and 10) also

provide an indication of the overall responsiveness of the system, since they show the initial and final errors as a function of latitude.

3. Section 2.2 and P5L20: In my understanding, inflation and localization are used in EnKF to address issues (such as filter divergence or long range spurious correlations) that result from under-sampling, i.e. having an ensemble so small in size that is not statistically representative of the state of the system.

Yes, this is our understanding as well.

I'd argue that this does not apply to the case of the "large" ensemble with 1518 members and it would also explain why for that ensemble size the WEP value does not significantly change as function of the localization length (figure 2).

Yes, this is likely the case.

The authors may want to comment on this and explain why and in what sense the large ensemble results that used no localization were worse than those with localization (P5L20).

The ensemble initialization technique we used involved sampling the truth run at regular time intervals. For the large ensembles, this required using a small time interval (36 minutes) and sampling late into the truth run when the dynamical variability of the system was settling down. Therefore, the ensemble members may not have been completely independent. Since the ensemble wasn't "perfect", applying some localization actually improved the system. We added a comment on this.

4. P5L4-6: I appreciate the reference to A15, but perhaps the authors can say something more here on the NNMI and make this manuscript more self-contained. Also in A15, instead of NNMI the authors used the acronym NMI. It would be good perhaps to either use a consistent acronym or acknowledge the use of a different name.

We will include some additional discussion of the normal mode initialization to make the paper more self-contained, and will also change to the acronym "NMI" rather than NNMI.

5. P7L11-12: I guess the wind vector RMSE is computed as in equation 6 of A15, perhaps a reference can be added.

This is nearly the same calculation. The only difference is that in the present paper we applied an area-weighting factor that we didn't apply in A15. We have included a comment to this effect.

6. Fig 7: You may want to specify in the right panel caption how the relative difference is computed.

We added a description to explain how the relative difference is computed.

I find interesting the reduced efficiency of the 50-member ensemble compared with those with 25 and 100 members. Is the reason for such a result known?

The reason for obtaining somewhat reduced efficiency with 50 members is unknown. This may be related to the fact that we tuned the system with one length scale for the assimilation of height and ozone (note that this feature doesn't occur for the height only assimilation). For unknown reasons, using a single localization length may have adversely affected the 50 member experiments more than the other experiments. In principle, we could simultaneously tune separate lengths for the two variables, since it is clear from newly numbered Fig. 3, and also discussion in A15, that the optimal length scale for ozone is smaller than for height. Why this would adversely affect the 50 member case is unclear, but it is a factor that is worth considering for future work.

**Response to RC2 from Thomas Milewski, 22 May 2016**

**Specific notes**

1) The choice of using streamfunction and velocity potential as control variables seems to be based on the experiments ran in A15 with the EnKF. What about the impact of choosing this set of control variables versus zonal and meridional winds in the 4Dvar system?

We haven't examined the impact of variable choice in the 4DVar system, but this would be an interesting follow up study. In the analytic algorithm that we used, based on Daley's formulation, the wind/wind and wind/geopotential height correlations are actually derived from streamfunction and velocity potential correlations using the *f*-plane approximation and isotropic assumption. So while the correlations are currently available for either set of variables, specification of the error standard deviations seems to be more straight-forward when using zonal and meridional wind. For example, we assume globally constant values for these quantities, which should be a reasonable assumption. It is less intuitive how to specify error standard deviations for streamfunction and velocity potential.

2) How did the authors come up with the value of 1518 as the number of dynamical state variables in the T21 system?

There are 253 modes for each variable used in spectral T21 system. This can be calculated by the following algorithm (in IDL notation). For given truncation NN, the total number of modes for a single global field is:

```
NN=21
totalmodes=0
for m=0,NN do begin
  nummodes=NN-abs(m)+1
  totalmodes=totalmodes+nummodes
endfor
```

Since there are 3 variables in the dynamical system, this results in $253 \times 3 = 759$ modes. And, since each mode is characterized by a complex amplitude, we multiply this by 2 to get the degrees of freedom for the dynamical system of 1518.

If I understand correctly, this is roughly the number of degrees of freedom in the dynamical system, but does it include the influence of O3?

Yes, the full system with O3 would have $253 \times 4 \times 2 = 2024$ degrees of freedom.

If not, this might explain why the optimal blending factor was not 1.0 for the large ensemble Z/O3 assimilation (before NNMI).

Yes, this may be a partial explanation of why the optimal blending is not 1.0, since our ensemble size is less than that of the entire system. We actually tried running the system with the size 2024 members. However, results were slightly worse than running with 1518 members. We think this may be due inadequacies in the ensemble initialization (see also response to RC1, question 3).

3) Optimal localization lengths for Z-only and Z/O3 assimilation are very different, suggesting that O3 error covariance structures and Z error covariance structures are probably different. Ideally, this should require separate localization lengths. Did you try tuning Z and O3 localization lengths separately?

We didn't try tuning the Z and O3 localization lengths separately, but that would be a logical follow-on to this project. Having an additional tuning parameter (2 length scales rather than 1) would undoubtedly improve the results, but would likely not change the overall conclusions. In addition, it would be interesting to explore tuning the system using correlation lengths that vary with latitude.

4) In figures 7, 8 and 9, the hybrid system is only compared to 4DVar, not EnKF. Since EnKF seems to outperform 4DVar for moderate to large ensemble sizes (at least in terms of WEP), it would be instructive to see the improvement that the hybrid system brings with respect to second-best performing system.

In the revision we added the EnKF results to the original Figs. 7, 8, and 9 (newly renumbered as Figs. 8, 9, and 10) along with some discussion.

5) Considering that this paper seems to complete a trilogy on the topic of wind extraction in a hierarchy of data assimilation systems, it would be interesting to have a final paragraph in the "Conclusions" section that is a more extensive review of the behavior of the different DA systems, possibly including the pros and cons of each.

This is an excellent idea. We decided to move much of the current conclusions in to a summary section and then used the conclusions section for a more extensive review.

**Technical notes**

1) P.6 L.16: Please correct "The WEP value for of Z/O3"

Done

[revised manuscript text omitted]